# Health-Monitoring Systems for Marine Structures: A Review

**DOI:** 10.3390/s23042099

**Published:** 2023-02-13

**Authors:** Arturo Silva-Campillo, Francisco Pérez-Arribas, Juan Carlos Suárez-Bermejo

**Affiliations:** 1Department of Naval Architecture, Shipbuilding and Ocean Engineering, Universidad Politécnica de Madrid (UPM), 28040 Madrid, Spain; 2Department of Material Science, Structural Materials Research Centre (CIME), Universidad Politécnica de Madrid (UPM), 28040 Madrid, Spain

**Keywords:** marine structural health, monitoring, sensors, ship structures, offshore platforms

## Abstract

This paper presents a comprehensive review of the state-of-the-art developments in health monitoring of marine structures. Monitoring the health of marine structures plays a key role in reducing the risk of structural failure. The authors establish the different sensors with their theoretical foundations and applications in order to determine the optimal position of the sensors on board. Once the data were collected, it was necessary to use for subsequent treatment; thus, the authors identified the different methodologies related to the treatment of data collected by the sensors. The authors provide a historical review of the location of different sensors depending on the type of ship and offshore platform. Finally, this review paper states the conclusions and future trends of this technology.

## 1. Introduction

Maritime transport is the main mode of international trade (up to 80% of goods); thus, marine structures are critical because of its specific influence to trade. Ships and offshore structures are used worldwide for a variety of functions and in a variety of water depths and environments. The correct selection of vessels and offshore platforms along with drilling methods, as well as the correct planning, design, fabrication, transportation, installation and commissioning of these offshore structures are hence very relevant [1].

Structural capability deteriorates with time as a result of damage accumulation, material wastage due to corrosion, and wear and tear [2]. The ability of the structure to maintain its original level of structural reliability over its service life can be significantly improved by several actions, e.g., corrosion control and structural monitoring plan in terms of inspections. Structural health monitoring (SHM) involves the observation and analysis of a component or system over time to identify the variation in any of its characteristics (physical, chemical or electrical), leading to a degradation of its present or future performance [3]. The incorporation of a health-monitoring system makes it possible to optimize the design, operation and/or maintenance, moving from criteria based on experience or conservative estimates, to others that take advantage of source of information about real-time in-service behavior. Structural monitoring also makes it possible to reveal the start of damage in structures that would otherwise remain invisible until a catastrophic/unexpected manifestation of damage occurs, allowing for decisions such as decommissioning for unscheduled maintenance actions or continuing until the next scheduled maintenance.

The International Maritime Organization (IMO) recommends the fitting of hull-stress-monitoring systems to facilitate the safe operation of ships. The use of this system will provide real-time information on the motions and global stresses that the ship experiences while navigating and during loading and unloading operations. The IMO recommendations are published in the Maritime Safety Committee circular, MSC/Circ.646 [4].

The main Classification Societies have included, in their normative review, different aspects related to the inclusion of the structural monitoring system. DNV GL uses HMON notation [5], which implies that an approved hull-monitoring system is to be installed on board. A hull-monitoring system may include sensors for several purposes, e.g., global structural response of the hull, local structural response of the hull, ice response monitoring, structural integrity management, environmental monitoring, seamanship, navigation, comfort, noise and vibration, rotating machinery, propulsion system, rudder performance, motions and accelerations, parametric roll, pressures, sloshing, and slamming.

Bureau Veritas uses the MON-HULL (Hull Stress and Motion Monitoring) notation [6], which is assigned to ships equipped with a hull-stress-monitoring system to provide real-time data of the ship on hull girder longitudinal stresses and vertical accelerations. The American Bureau of Shipping uses HM (Hull Monitoring) notation [7], which is classified as follows: HM1 for slam warning, ship motion, green seas warning, sloshing monitoring and sea state; HM2 for local load, hull girder stress, fatigue monitor and structural temperature monitoring; HM3 for voyage data recorder (VDR), navigation, wind and shaft monitoring.

Due to the importance of correct implementation of the on-board monitoring system of marine structures, several projects financed by the European Commission have concentrated their efforts on these aspects in order to optimize and improve the on-board structural monitoring systems. HULLMON+ [8] is a monitoring tool used to control the motions and stresses state of a marine structure in such a way that reduces the potential for structural failure due to severe weather, fatigue and pollution, improves passenger comfort, and minimizes loss of life and damage cargo on board.

The main purpose of the SHIP INSPECTOR project [9] is to implement long-range ultrasonic test procedures to detect defects in the hull structures of ships. Another project was the OPTIMUS [10], whose main objective is to develop a mathematical algorithm for optimal placement of strain sensors (strain gauges, fiber Bragg grating or other) in composite structures for maximum damage detectability. A relevant project in the aspect of structural monitoring on board was MONITAS (Monitoring Advisory System) JIP (Joint Industry Project) [11], which provides a description of the system developed, installed and in operation on board the FPSO Glas Dowr in the Sable field. The monitoring system is driven by fatigue life consumption from fatigue design data and global and local loads.

The information provided by the sensors that are used to monitor structural health can be processed under different perspectives that improve the efficiency and reliability of the structures. In this aspect, different alternatives have been developed, such as the technique known as Digital Twin, which is the digital reproduction of a physical structure, describing its characteristics and properties including the acquisition, processing and representation of sensor data. VanDerHorn et al. [12] worked on a Digital Twin concept for monitoring the fatigue damage on ships based on wave condition.

Another processing technique is known as Internet of Things (IoT), which is a network of digital interconnectivity between the Internet itself, people and devices that enables the exchange of data between them. IoT allows for key information about the use and performance of devices and objects to identify patterns, provides recommendations and improves performance. Different authors have mentioned this topic in their research, such as Wu et al. [13], where they discussed the main types of external loading of offshore platform health-monitoring technology by means of a satellite communication system and a wireless sensor network.

One of the recent advances in the technique of processing the information provided by sensors located in the marine structures is known as machine learning [14], which is a data analysis method that automatizes the development of analytical programs. This technique is often based on artificial neural networks (ANNs). It is a field of artificial intelligence that is based on the concept that systems can learn from data, make decisions with minimal human intervention and identify patterns. Different research studies reflect the importance of this aspect in structural health monitoring. Kawai et al. [15] estimated the encountered sea state using machine learning from measurement data of ocean-going 14,000 TEU container ships. Karvelis et al. [16] reported a novel data-driven method for the localization of acoustic emissions in complex structures of ship hulls, combining insights from the fields of signal processing. Jiang et al. [17] proposed an innovative pose-relay video metrics measurement system for ship deformation.

The use of sensors for structural monitoring of marine structures is independent of the material where they have to be placed, both in steel or in a composite material. There has been a lot of research related to the feasibility of the arrangement of sensors in this last type of material; Mieloszyk et al. [18] presented an application of embedded FBG sensor arrays for evaluation of a complex composite structure in a fast patrol boat. Silva-Muñoz et al. [19] introduced an approach for monitoring composite joints using strain data measurements by distributing and integrating fiber Bragg grating (FBG) sensors. Li et al. [20] established a monitoring approach for composite marine joint structures on the basis of strain readings under operative loading using embedded FBG sensors. Sante and Donati [21] conducted a research study into several approaches for efficient and reliable embedded FBG sensors in a composite material, with specific application to the manufacturing process of sailboat masts using the bagging technique.

This paper provides an in-depth review of the different aspects involved in the structural health monitoring of marine structures. The main novelty of this paper is that there is not such a comprehensive review of sensors in marine structures in the current literature. In Section 2, the authors establish the existing technology within each of the types of sensors that may be available on board, with their theoretical foundations, applications, and their advantages and disadvantages. In Section 3, the authors determine the different current theories to establish the optimal position of the sensors on board. In Section 4, the authors review the different methodologies related to the processing of the data collected by the sensors. In Section 5, the authors perform a historical review of the different sensors located depending on the type of ship and offshore platform. Finally, in Section 6, the authors give the conclusions and indicate the future trends of this present and future technology.

## 2. Sensor Technology

### 2.1. Electrical Strain Gauge

An electrical strain gauge is an electronic instrument used to measure deformations due to stresses applied to an object. It was invented by Edward E. Simmons and Arthur C. Ruge in 1938 [22] and its operation is based on the variation of the electrical resistance of a gauge as a function of the strain undergone by the instrument. To measure such changes in resistance, strain gauges are usually used in a Wheatstone bridge configuration with a voltage excitation source. The Wheatstone bridge allows the change in resistance to be translated into a voltage output [23,24]. Short baseline (SG) gauges are great for measuring shear stresses, combined with their low cost and capability to be installed in confined locations. Long baseline strain gauges (LBSG) are used to measure hull girder stresses (global strain) on commercial vessels. Long baseline strain gauges usually contain a long rod rigidly connected at one end to the hull. The second end is free to move through a guide assembly that ensures axial motion only. The amount of free end movement (measured from a set of “zero” deflection points) divided by the rod length gives the average deflection in the shell.

Hull girder stress measurements are associated with the wave and still water bending moments, both acting on the vessel. A strain range of −2000 to +2000 microstrains should be expected for steel vessels. The measurement uncertainty should be less than +/− 20 microstrains or +/− 5% of the measurement reading, whichever is greater. A typical bandwidth should be 0–1 Hz. The measurements of hull girder stresses are generally performed with multiple LBSGs arranged along the length of the vessel. Strain gauges to monitor hull girder stresses should be positioned as close as possible to the places where the ship’s loading manual and the loading instruments provide the bending moment results. Where strain gauges are placed in locations under multiple loading mechanisms, means shall be available to separate the individual stress components.

The minimum required number of strain gauges and their approximate position for tankers, bulk carriers, general cargo ships and container ships are given below: two amidships (one on port and one on starboard side on deck), one at 25% of the length from the bow (on deck) and one at 25% of the length from the stern (on deck).

### 2.2. Fiber Optic

#### 2.2.1. Point Sensing

Fiber Bragg gratings are created based on the photorefractive technique. The bare fiber is excited in a hydrogen atmosphere and then printed using an ultraviolet laser. The printing is performed by one of many methods and leaves a series of identically interspaced stripes across a region of fiber [25,26,27,28,29]. This array of stripes is called a Bragg grating; the stripes are effectively very short areas that have a very small, slightly varying index of refraction. In a fiber that has a Bragg grating, the reflected broadband light is transmitted back to the light source at a particular frequency that matches the wavelength of the grating. The rest of the light frequencies are not affected by the Bragg grating.

##### Long Period Grating (LPG)

The LPG has a period typically in the range of 100 µm to 1 mm, and this technology promotes coupling between the propagating core mode and co-propagating cladding modes [30,31,32,33,34].

##### Interferometric Cavities Sensors

A fiber optic interferometer operates on the interference effect between two beams that propagate through different optical paths of a single fiber or two different ones [35,36,37,38]. Interferometric sensors are classified as Mach–Zehnder Interferometer (MZI) [39,40,41], Fabry–Perot Interferometer (FPI) [39,41], Michelson Interferometer (MI) [40,42], and Sagnac Interferometer (SI) [41,43,44].

##### Photonic Crystal Fibers (PFC)

PCFs are commonly applied as a single optical material to fabricate sensors by air holes that may be filled with various materials (liquid, gas, or even solid) to change the difference in refractive index, and they have been applied in several fields of sensor search [45,46,47,48]. They can be grouped as index-guiding PCFs [49] and photonic bandgap PCFs [50].

##### Surface Plasmon Resonance (SPR)

SPR is the basis for achieving high efficiency. The resonance status at the barrier interface of a metal thin film and a dielectric medium can be coupled in an optimal way by p-polarized light at a given wavelength and angle of incidence. It is produced when the value of the surface plasmon wave vector is equal to the constant of propagation of the incident light [51,52,53,54].

#### 2.2.2. Distributed Sensing

There are two main types of distributed optical fiber sensors: optical frequency-domain reflectometry (OFDR) [55,56] and optical time-domain reflectometry (OTDR) [57,58,59]. There are different technologies: Raman scattering [60], Brillouin backscattering [61] and Rayleigh backscattering [62,63].

#### 2.2.3. Ultrasonic-Guided Waves (Lamb Waves)

A Lamb wave represents a guided wave (GW) generated in a plate-shell assembly. Lamb waves have two basic modes: symmetric and antisymmetric. Damage prediction systems and failure prediction techniques have recently been the subject of intensive research and development in a wide range of engineering applications such as underwater pipeline monitoring [64].

### 2.3. Acoustic Emission (AE)

The monitoring of AE is the transitory acoustic stress wave, which results from material changes. The main principle of AE consists of using an array of sensors to find characteristic sound signals that may indicate the existence of structural defects in the material. AE has many applications, although the most relevant one for offshore platforms is the structural monitoring of failure mechanisms such as cracks. The method has been applied in locations where a high potential for fatigue cracking is known and where inspection is technically difficult or expensive. The monitoring system can accurately detect crack initiation and growth and may be combined with strain gauges to provide a correlation between AE readings and structural stress levels. AE as the system allows for real-time feedback on fatigue crack initiation and growth; it may be used to identify fatigue failure at an early phase before it is detected by standard NDT techniques. Due to the AE signal attenuation, this approach is appropriate for monitoring locally within a few meters of the structure. The AE sensors may be placed outside or inside the structure [65].

### 2.4. Fatigue Damage Sensor (FDS)

FDS is a sensor type gauge, which is bonded to the steel structure and shows the fatigue damage magnitude under stress. The FDS is built with two thin metal sheets. A fatigue crack starts in the FDS when it suffers a certain amount of stress duration of variable magnitude. The fatigue crack length is transformed to structural fatigue damage, which is based on the crack growth characteristics of the thin metal sheets [66,67]. There are multiple companies with fatigue damage sensors; Kawasaki Heavy Industries, Ltd. developed a robust, compact and extremely fatigue-sensitive sensor, which can be coupled for fatigue damage monitoring in the proximity of weldment members of structures subjected to fluctuating loads. In FDS, the crack will be propagated from the initial notch after receipt of fluctuating stresses in the structural elements. Fatigue damage may be monitored by measurement of the crack length in FDS by using the key feature of FDS: the crack growth rate is constant and always independent of the crack length in the same strain fluctuation ranges. The sensor consists of a base and a sensing foil. A groove is formed in the center of the sensing foil to amplify the strain, and an initial notch is placed in the center of this groove.

Fatigue damage estimation is obtained by the measurement of the crack length in the sensor. It is composed of a sensing foil that is thinner in the middle section to increase the strain amplification. An initial notch is introduced in the central part of this area. When the FDS is fixed to a structure, it suffers the same deformation variations as the structure and thus initiates a fatigue crack that is propagated from the tip of the notch as a function of the number and degree of the stress cycles. The crack propagation rate is considered to be relatively independent of the length of the crack. The crack length is proportional to the fatigue damage accumulated in the structure over the time after sensor installation; therefore, the crack length is inversely proportional to the fatigue life consumed.

Another FDS is the CrackFirst^TM^ fatigue control, designed for application on welded steel structures. It is a thin wedge of material with a fabricated pre-crack in its center, which is clamped to the target structure close to a crucial location. Under the cyclic stress action, the pre-crack is extended in a proportion to the cumulative fatigue damage for a welded connection under the same loading. The sensor status shows the quantity of design life consumed in the weld adjacent to the joint. This is a new type of passive fatigue damage sensor that has been developed for fatigue-critical locations in pinned, welded or other stress-concentrated joints in steel structures.

A new intelligent RFID (radio frequency identification) FDS has been developed and patented for the prediction of residual fatigue strength of mechanical and structural critical components in structural health monitoring [68]. The proposed smart sensor system is designed for early detection and estimation of the structural health cumulative fatigue damage level, and it wirelessly transfers the information using an active or passive RFID integrated system. The developed RFID fatigue sensor system has a specially designed geometry with multiple parallel oriented unidirectional, bidirectional or multidirectional breakable C-, U- or V-type notched beams, having different fatigue lifetimes to predict unidirectional or bidirectional fatigue damage level of structural or mechanical elements, including composite structures [69].

### 2.5. Accelerometer (ACC)

Ship motions are usually measured with accelerometers. The different accelerometer types are supported by different physical principles (uniaxial and triaxial) and are appropriate for different measurement applications. There are two types of accelerometers: the AC (alternating current) response accelerometer, which is only one able to measure acceleration changes (but not a constant acceleration), and the DC (direct current) response accelerometer, which can measure both acceleration changes and a constant acceleration. Table 1 shows a comparison of the most commonly used types of accelerometers and their possible applications [70].

Küchler et al. [73] presented an observer-based method to estimating the heave motion of a vessel from accelerometer signals without the need for vessel-specific input parameters. The motion reference unit (MRU) is a motion sensor that can measure roll, pitch, heave, accelerations, angular velocities, and velocity of any platform, ship, or vessel on which it is mounted [74,75]. The motion reference units include accelerometers and gyroscopes that are able to detect motions in space without following any external objects. Acceleration measurements at the bow are linked to the vertical motion (heave and pitch) of the vessel and the first mode of the vertical vibration of the hull girder. Depending on the ship size, an acceleration range of −20 to +20 m/s^2^ should be adopted. The uncertainty of measurements should be less than +/− 0.2 m/s^2^ or +/− 5% of the lecture, whichever is greater. The typical bandwidth should be 0.02 to 1.0 Hz [76].

### 2.6. Pressure Transducer (PT)

A PT is a system that indicates the pressure of a fluid, showing the force on the surfaces in contact with it. PTs are commonly applied in reservoirs, storage tanks, hydraulic systems, cooling systems and flood alarms. These sensors are built to operate for millions of cycles and are intended for submersion. Three technologies are essential to the advanced performance of a modern transducer: application-specific integrated circuit (ASIC) electronic packaging, chemical vapor deposition (CVD) technology and thin-film sputtering technology. The combined effect of these mechanical and electronic developments has resulted in a growing interest in their application in the marine industry [77]. ESI^®^ Technology is specialized in the development and fabrication of pressure transducers for use in the marine industry. Table 2 shows the physical variables as a function of sensor type.

## 3. Optimal Positioning of Sensors

The basic idea of optimal sensor positioning applied to the continuous monitoring of structures is to identify their optimal configuration in the structure so that they can capture as much information as possible about the dynamic behavior. That is, sensors should be able to pick up the desired number of vibration modes and distinguish one from the other correctly, as well as obtain the natural frequencies of each of these modes in order to perform an effective monitoring of the structures. The sensors must be placed in those positions of the structure where the response of the vibration modes is greatest. A priori, this may seem simple, but when the sensors are required to obtain the necessary information from a large number of modes, it complicates the resolution of the problem. The higher the frequency associated with a mode, the more complex its shape. The greater the number of zero-acceleration nodes it has, the greater the number of modes to be obtained by the sensors and the greater the number of nodes present in the optimal positioning of sensors (OPS) analysis of the structure. This makes the optimal placement of sensors difficult, since in order to place the sensors in the optimal positions, it is necessary to avoid positions where the nodes of the vibration modes are located, since in these positions, the information of the mode is lost, as its response is not captured.

There are several ways to choose the best locations for the sensors. Mathematically, optimal sensor locations can be formulated as a constrained optimization problem in which the variables are the candidate sensor locations on a structure. The constraints are basically determined by the available degrees of freedom (DOFs) and the total number of available sensors. The objective function to be minimized or maximized is defined to measure the utility of a sensor configuration in optimal sensor locations based on different criteria.

### 3.1. Inverse Finite Element Method (iFEM)

Tessler and Spangler [78,79] developed a methodology called the inverse finite element method (iFEM), which certainly meets the needs of the SHM procedure. The iFEM approach calculates the structural deformations from the experimentally measured deformations, on the basis of the minimization of a weighted least-squares functional. The iFEM approach has wide applicability to challenging structures under complicated boundary conditions in a real-time environment. The iFEM framework is fast, powerful and accurate enough for real-time implementation of all types of static and dynamic loads and a wide variety of elastic materials, because only the deformation–displacement ratio is considered in the formulation [80,81].

Kefal and Oterkus [82] formulated a four-node inverse shell element (iQS4) that includes degrees of freedom of hierarchical drilling rotation for numerical simulations. The application of iFEM to SHM of marine structures is developed employing several kinds of low-fidelity and high-fidelity discretization techniques for the given conditions. The impact of sensor placement, its number and the discretization of the geometry on the accuracy of the solution is examined. Kefal et al. [83] developed an intelligent system that determines the most convenient locations of on-board deformation sensors for SHM of marine structures applied to a long barge.

Kefal and Oterkus [84] tracked the displacement and stress at the midship of a chemical tanker using the iFEM approach. The numerical performance of the iFEM procedure is performed by using a four-node inverse quadrilateral element (iQS4). It has been found that the employment of the deformation data collected from the deformation rosettes located on the central longitudinal bulkhead, central deck stiffener and the central girder can be enough to reproduce a precise global deformed mode and the von Mises stresses that are caused by the shear force and vertical bending moment due to the oscillatory motions. Therefore, it can be suggested that these elements are the optimal placement for sensors on board. Kefal and Oterkus [85,86] studied in the same regime for beam seas to calculate the horizontal and vertical wave bending moments and the wave torsional ones, which act on the parallel midship body of a container ship.

Kefal et al. [87] found a new inverse triangular shell element (i3-RZT) based on the improved iFEM formulation for real-time structural health monitoring of laminated and sandwich structures. The kinematic field of the i3-RZT element supports quadratic interpolations that allow for a robust implementation of drilling DOF that has the benefit of preventing singular solutions when modeling complex shell structures. Kefal and Oterkus [88] particularized the same study for thin-walled stiffened steel cylinders used as compression elements in the field of offshore structures with a damaged local. Kefal et al. [89] and Oterkus et al. [90] analyzed by hydrodynamic and finite element modeling the FBG sensor deformation data of a capsized bulk carrier floating in bow seas conditions, using iFEM/iQS4.

Kefal [91] developed a new eight-node curved inverse-shell element, named as iCS8, based on iFEM methodology for different cylindrical marine structures subjected to static and dynamic loads. Li et al. [92] demonstrated the applicability of iFEM to SHM of submarines focused on sensor placement on a standard submarine pressure hull. Li et al. [93] applied the inverse finite element method (iFEM) to monitor the tower of an offshore wind turbine under both static and dynamic loading conditions. The total displacements and von Mises stresses obtained from iFEM analysis are compared against literature results, and optimum sensors location are determined.

Ghasemzadeh and Kefal [94] coupled the genetic algorithm (GA) method with iFEM to optimize the location of the remaining sensors. Ghasemzadeh et al. [95] utilized the iFEM to scrutinize and detect structural damage in terms of corrosion pits in marine structures.

### 3.2. Fisher Information Matrix (FIM)

Fisher information is a way of quantifying the amount of information relative to an unknown parameter contained in a distribution modeled by an observable random variable. Formally, it is the expectation of the observed information. In cases with several parameters, a Fisher information matrix is defined, used in statistics to calculate covariance matrices associated with maximum likelihood estimates [96]. Yao et al. [97] used a genetic algorithm to place sensors optimally on a large space structure for the purpose of modal identification. Guo et al. [98] presented a sensor placement optimization based on damage detection and some improved strategies in genetic algorithms optimization for sensor location. Lian et al. [99] proposed a new methodology to select the best sensor locations for large structures. This method maximizes the contribution of each sensor to modal observability and simultaneously avoids the redundancy of information between the selected degrees of freedom.

### 3.3. Information Entropy (IE)

Information entropy is known as the unique measure of the probabilistic uncertainty of the parameters of a model. It was first defined in 1949 [100] and is also known as Shannon’s entropy due to its authorship. The IE depends on the determinant of the FIM and justifies the use of the determinant instead of the matrix trace, as other Fisher matrix approaches do. It is used in the process of sensor configuration optimization, as well as to calculate dynamic characteristics for nonlinear model identification. Yi et al. [101,102,103] proposed a novel methodology called asynchronous-climb monkey algorithm (AMA) for the optimum design of sensor arrays for a structural health monitoring system.

### 3.4. Modal Assurance Criterion (MAC)

The modal assurance criterion is based on measuring the correlation between two vibration modes, which can be obtained from different sources (finite element model, sensors on the structure, etc.). Yuen et al. [104] presented a methodology to design optimal and cost-effective sensor locations and configurations for updating structural models and health monitoring, supported by an informative entropy estimation of the uncertainty in the model value obtained by a statistical approach to system recognition. Chow et al. [105] introduced an approach for identifying the optimal locations to place a number of sensors within a structure in such a way to obtain as much data as possible to update the structural model.

## 4. Data Processing

The data processing allows a statistical data set to be recorded for off-line operation and real-time values to be displayed for on-line operation [106], and the recording duration per cycle shall be adjusted to provide results that do not vary by more than 10% from one wave encounter to the following wave event under steady-state condition of navigation [6]. Data processing methodologies can be described as statistical, probabilistic, knowledge-based, reasoning methods and inference [107]. Probabilistic approaches include Bayesian networks, maximum likelihood estimation methods, inference theory and Kalman filtering. Statistical approaches include covariance, cross-variance and other statistical analyses [108]. Knowledge-based methodologies involve genetic algorithms, fuzzy logic and artificial neural networks [109,110]. There are other methods widely used, such as fatigue rainflow cycle counting for fatigue life evaluation, Fourier transform (FT/FFT) for modal analysis, short time Fourier transform (STFT) for modal analysis and wavelet transform (WT) for modal analysis.

The structural reliability problem involves two opposing quantities: a capacity (or resistance, supply, strength, etc.) and a demand (or load, stress, load effect, etc.). The structure is said to have failed when the demand (*S*) exceeds the capacity (*R*), with *M = R − S*. The probability of failure is defined by
(1)$$p_{f}=P\left[M\leq 0\right]≃\Phi \left(-\beta \right)$$

The complement of the failure probability, 1-*p_f_,* is called the reliability, where Φ is the standard normal distribution function. The updated probability of failure can be determined by using the following conditional probability:(2)$$p_{f}=P\left[M\leq 0|E\right]=\frac{P\left[\left(M\leq 0\right)\cap E\right]}{P\left[E\right]}$$
where *E* is the possible result from inspection event [111]. Kelangath et al. [112] introduced the use of data-driven Bayesian modeling in risk analysis and made a comparison with the different data-driven Bayesian methods available for a better and safer shipping practice. Yuen et al. [113] proposed a Bayesian algorithm for sequential sensor placement because of robust information entropy for multiple types of sensors. Xiao et al. [114] presented a global framework for the analysis of the reliability of local ship structures, based on Bayesian data merging. Stull et al. [115] reported the results of an active ongoing project to employ techniques for model-based structural health monitoring on novel and current hull structures, based on an optimization-based and a Bayesian approach. Zhu and Frangopol [116] presented an approach to provide improved accuracy in the evaluation of ship cross-section reliability and redundancy by employing the Bayesian method. Decò and Frangopol [117] developed and proposed a risk-based framework approach for marine structures that includes information from SHM [118]. Their proposal is based on a simulation-based Bayesian update approach and the application of the Rayleigh distribution.

## 5. Monitoring Systems in Shipping and Offshore Industry

### 5.1. Ship Structures

Table 3 depicts the typical sensors located on board for the common commercial vessel types.

#### 5.1.1. Container Ship

Yu et al. [119] presented a full-scale measurement solution designed to monitor the ship motions, wave conditions and structural response on an 8063 TEU container ship. Nielsen et al. [120] obtained a calculation method for the prediction of fatigue damage ratio in hull structures considering whipping stresses, by means of full-scale wave-induced stress ranges in a container ship in which the related fatigue damage ratios are calculated. Chen et al. [121] investigated a new approach to calculate the directional wave spectra from the measured ship responses in different scenarios of a large 10,000 TEU container ship under the combination of different stress and movement sensors.

Chen et al. [122] extracted data from the full-scale measurements of a 14,000 TEU double bottom container ship, whose sea states are similar to those of numerical simulation among the existing time series, in order to analyze the correlation between the double bottom and hull girder bending, on wave and vibrational response. Miyashita et al. [123] presented full-scale calculations (longitudinal stresses) of an 8600 TEU container ship carried out during four years and two months. The longitudinal stresses measured through sensors located on the midship are divided into horizontal bending stresses, hull girder vertical bending stresses, axial stresses and warping stresses. Chen et al. [124] discussed an approach, based on wave spectra that are estimated from restricted measurement data, to reproduce unmeasured hull stress responses.

#### 5.1.2. Bulk Carrier

Kim et al. [125] showed the typical general arrangement of a bulk carrier with the common location of sensors on this type of vessel.

#### 5.1.3. Oil Tankers

Several monitoring installations were carried out on ships in 1990 [126] and on other ships between 1990 and 1991 [127]. This system consists of about six long deformation sensors, two pressure sensors, an accelerometer and motion sensors. Hu and Prusty [128] found that in order to obtain more data on the deformation of ship hulls, it is necessary to deploy more strain gauges on the side hulls and transverse bulkhead. In terms of the dynamic loads of the ship, it is necessary to install pressure sensors at different locations on the side hulls. Accelerometers need to be mounted in three directions to measure the pressure of cargo oil.

#### 5.1.4. Others

Stull et al. [115] reported the results of an active research project aimed at employing pattern-based structural health monitoring approaches on emerging and current hull structures. Takaoka et al. [129] focused on the implementation of FDS on FPSOs for fatigue life consumption. Zhang et al. [130] provided a statistical and artificial intelligence approach to conduct resilient condition monitoring applied to a hawsers system in a FPSO oil offloading system. Kaminski [131] described the data processing and interpretation tools to prevent the fatigue damage of FPSOs. Thomas et al. [132] studied the application of extended stress, motion and hull wave measurements, together with a sophisticated finite element model, under asymmetric loading case.

Sato [24] conducted a fatigue monitoring program on a 135,000 m3 LNG carrier consisting of a comprehensive range of measurement elements. Yamamoto et al. [133] distributed a compact fatigue damage sensor into LNG carrier that can identify accumulated fatigue damage in welded structures by mounting it to the component and inspecting it after a certain time. Drummen et al. [134] presented the setup of two types of hull-structure-monitoring (HSM) systems. One is an overall system focused on global load effects and strains. The other is a local system aimed at detecting cracks through acoustic emission (AE) monitoring. Long base strain gauges are an important part of the global HSM system that was installed on USCGC STRATTON. The location of these sensors was optimized using hydro-structural calculation tools [135].

Cusano et al. [136] described and compared both full-scale and model-test monitoring campaigns and the most significant collected data, referring to the occurrence of slamming events on the MDV−3000 mono-hull ro-ro passenger fast ferry, focusing on the longitudinal bending moment amidships. Jensen et al. [137] studied a fast patrol boat (KNM Skjold), a twin-hull surface effect ship (SES) made of fiber-reinforcement polymer (FRP) sandwich composites with 47 m long and 13.5 m wide through the Composite Hull Embedded Sensor System (CHESS) project, of which the main objective was to design, develop, fabricate and install strain-monitoring systems using distributed fiber optic sensors for hull structure monitoring. This paper presented a method for measuring the global loads based on extensive finite-element analyses and strain measurements from networks of FO strain Bragg sensors attached to the hull of KNM Skjold.

Torkildsen et al. [138] presented a thorough introduction to the ship hull structural health-monitoring system, with the analysis of the data recorded onboard the Royal Norwegian Navy (RNoN) Mine Counter Measure Vessel (MCMV) “HNoMS Otra”. The HSV-2 Swift (HSV 2) is a high-speed vessel (wave piercing catamaran) of the United States Navy. Sielski [139] studied this vessel for detecting damage in the structure and to extend this detection capability to a prognosis capability [140]. Another investigation that also deals with the HSV-2 vessel is the one proposed by Mondoro et al. [141], where they provided a method for predicting the responses of vessels in non-observed cells by integrating data from the observed limited number of cells.

Majewska et al. [142] presented and discussed experimental research on tall sailing ships using FBG sensors for the foremast. They determined the stress/strain level of the foremast during her normal operation to establish the effectiveness, quantity and configuration of the sails for the strain/stress level of the foremast. Hageman et al. (2015) [143] conducted a hull fatigue monitoring system for FPSO by coupling a minimum sensor array with automated data planning processing to collect the hull response, fatigue loading and wave conditions encountered. Ferreira et al. (2017) [144] described a fiber optic strain gauge sensor-based monitoring system used to recreate in real time the form of a sail using an algorithm developed and integrated to convert strain readings into deformations.

Söder et al. [145] presented a method for monitoring the stresses in ro-ro vessels by real-time measurement of ship motions. Johnson et al. [146] proposed a wireless, rapidly deployable hull monitoring system with an associated analytical framework that employs hull measurements to evaluate the lifecycle performance of a ship.

Yan et al. [147] suggested a new technology for the monitoring of the structural health of the stinger of a large deep-water pipe-laying vessel. Roberts et al. [148] presented field tests on the 138 m passenger and vehicle ferry Smyril, operating in the North Atlantic Ocean, using GPS and FBG kinematic sensors. Hageman and Thompson [149] assessed, through a virtual monitoring technique, the structural behavior of a frigate ship. There is a multi-purpose cargo vessel (INF 2 classified) named Atlantic Osprey with four fatigue damage sensors on board [150].

### 5.2. Offshore Structures

The monitoring of offshore structures can be classified as metocean factors related with the environmental conditions (wind, waves, currents, ice, etc.) and structural operational status [151,152].

#### 5.2.1. Tension Leg Platform (TLP)

Van Dijk and van de Boom [153] evaluated, by means of full-scale monitoring, the Marco Polo Tension Leg Platform under exposure to hurricane and loop-current conditions, under high- and low-frequency modes of motion, under the fatigue loading of the platform, and under the dynamic behavior of the tendons and risers with focus on the vortex-induced vibrations.

#### 5.2.2. Wind Turbine

Mieloszyc and Ostachowicz [154] presented an application of the SHM system based on FBG sensors dedicated to an offshore wind turbine support structure (tripod) model. Nejad and Moan [155] studied the health monitoring of a 5 MW spar wind turbine drivetrain by means of a decoupled analysis method. Kim et al. [156] suggested and tested a structural health-monitoring method for floating offshore wind turbines (FOWTs) by using modal analysis with signals from numerical sensors. Kou et al. [157] studied, for a semi-submersible platform under repair, eight old main supports that connect the columns to the pontoon in order to substitute them with new ones, by means of a structural stress monitoring of eight key points calibrated with finite element models.

Moreira and Guedes Soares [158] found a technique using artificial neural networks to estimate wave-induced vertical bending moment and shear force from ship movements to incorporate it into a hull monitoring system. Vidal et al. [159] proposed a methodology for the detection and localization of damage in a wind turbine with jacket foundations, using eight triaxial accelerometers to identify any anomaly in the structure’s dynamic behavior. Yang et al. [160] proposed an offshore wind turbine deck-structure-monitoring system consisting of vibration, deformation and corrosion.

#### 5.2.3. Jacket Platform

Lotfollahi-Yaghin et al. [161] conducted a numerical investigation in a jacket offshore platform that operates in 70.2 m water depth in the Persian Gulf. Sun et al. [162] studied the dynamic response of a jacket offshore platform under a seismic excitation model based on the collection of data provided by the FBG sensors and strain gauges. Ge et al. [163] examined the stress response of the jacket legs by FBG sensors to determine the impact strength on the same. Ali et al. [164] provided an experimental approach using piezoelectric sensor detection and finite element analysis method for the analysis of fatigue cracks in three types of joints. Tang et al. [165] proposed the methods of structural monitoring and early warning status on the basis of the aged coating characteristics of offshore platforms. Liu et al. [166] detected damage to the structure of the jacket platform by analyzing the acoustic emissions.

#### 5.2.4. Jack up Platform

Archer [167] studied the monitoring of cyclic loads on a leg of the jack-up platform Nengue Sika during a transport from Singapore to West Africa. Shabakhty et al. [111] showed, based on a crack propagation approach and achieved information from inspection, that the remaining fatigue reliability of jack-up structures could be determined and updated by using a Bayesian procedure in the duration of the service time. 

#### 5.2.5. Spar

The dynamic response of the Neptune Spar platform was monitored under hurricane sea conditions [168]. Thethi et al. [169] presented a riser-monitoring strategy and implementation on a deepwater Gulf of Mexico Spar top tensioned riser. Karayaka et al. [170] installed riser and flowline monitoring (RFM) on Chevron Tahity Spar to study the dynamic response of the catenary [171].

## 6. Conclusions and Future Trend

This paper provides an in-depth review of the different aspects involved in the structural health monitoring of marine structures. This investigation aims to be of help when selecting a structural monitoring system on board, depending on the characteristics of the ship or offshore structure, and the measurements to obtain. The authors have shown that the health monitoring of marine structures is a profoundly interesting area due to the information it provides for the user in the prevention of structural failure. The authors have highlighted the importance of the configuration of such sensors by comparing different theories, highlighting above the others a recent methodology based on the inverse finite element method (iFEM), due to its generalist and easy-to-use nature. Through this review, the reader is provided with the existing technology within each of the types of sensors that may be available on board, with their advantages and disadvantages.

It has to be emphasized the great importance of having, whatever the type of vessel or offshore structure, a structural monitoring system that allows for knowing in real time the structural health, with the great advantages that derive from it. As a future trend, a great influence of FBG technology can be observed, due to the substantial improvement in terms of reliability that can generate and build the Digital Twin based on the IoT, where the current industry is focusing efforts.

## Figures and Tables

**Table 1 sensors-23-02099-t001:** Commonly available accelerometer types [71,72].

Technology	Response	Pros	Cons	Application
Capacitive Micro-Electro-Mechanical Systems (MEMS)	DC	Inexpensive, small size and easy to integrate into electrical systems	Poor signal-to-noise ratio, limited bandwidth	Applicable to estimate displacement and velocity through integration over time.
Piezoresistive (PR)	DC	Accurately calculating velocity or displacement	Low sensitivity, temperature compensation	Applicable to estimate displacement and velocity through integral over time
Charge mode piezoelectric (PE)	AC	Good sensitivity, easy installation, durable in hostile environments	Need special cabling to shield from noise, requires a charge amplifier	Could be applied to extreme temperature conditions
Voltage mode Internal Electronic Piezoelectric (IEPE)	AC	Good sensitivity, easy installation, low noise levels, easily integrated with other systems	Ability to tolerate hostile environments when compared to PE accelerometers	Large operating temperature range

**Table 2 sensors-23-02099-t002:** Common physical variables and sensor types [70].

Physical Variable	Recommended Sensor Types
Hull bending moment (vertical, horizontal and torsional in longitudinal direction)	LBGS
Sectional force (vertical and horizontal shear force)	LBGS or SG (electrical or fiber optic type)
Slamming event detection	ACC or PT
Pressure (slamming, sloshing, wave, etc.)	SG (electrical or fiber optic type) or PT
Rigid body motion (6 degrees of freedom)	MRU
Vibration—low frequency	ACC
Vibration—high frequency	ACC
Structural deflection	ACC or SG (electrical or fiber optic type)
Local strain/stress	SG (electrical or fiber optic type)
Structural modal shapes and natural frequency	SG or ACC
Crack initiation and propagation	AE or FDS

**Table 3 sensors-23-02099-t003:** Structural sensor for common commercial vessel types [70].

Direct Measurand	Container Ship	Bulk Carrier	Oil Tanker	Ro-Ro Ship	LNG	Passenger Ship
Vertical accelerations at bow	ACC (*)					
Transverse acceleration amidships	ACC (*)		-	ACC (*)	-	ACC (*)
Ship motion (at center of gravity)	MRU (**)	MRU (*)	MRU (*)	MRU (**)	MRU (*)	MRU (*)
Global longitudinal stress amidships (port and starboard side)	LSBG (**)					
Global longitudinal stress at quarter length fore and aft of midship (port or starboard side)	LSBG (**)			LSBG (*)		
Local transverse stress at transverse deck strip amidships	SG (**)	-				
Global longitudinal stress below neutral axis amidships (port and starboard)	LSBG (*)		-			
Double bottom bending stress	SG (*)		-			
Lateral loads at bowflare or bottom near forward perpendicular (Slamming pressure)	PT/SG (*)					
Lateral loads at side shell (wave pressure)	PT/SG (*)					
Sloshing response of liquid in tanks	-		PT/SG (*)	-	PT/SG (*)	-

(*) optional/recommended, (**) typically required.

## Data Availability

All data are presented in the paper.

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
