# Peer review of "Health-Monitoring Systems for Marine Structures: A Review"

_sensors, 2023, doi:10.3390/s23042099_

Round 1

Reviewer 1 Report

This paper presents the review of the state-of-the-art development in SHM for marine structures. Sensor technologies, location, and data processing methods are discussed. The monitoring systems in the shipping and offshore industry are accordingly presented to show the innovation.  However, there remain some problems that need to be fixed to improve the manuscript:

1.      The author claimed that the main novelty of this paper is a comprehensive review of sensors in marine structures. So, the main problem is that the authors should focus more on the difference between SHM techniques for normal infrastructures and the marine structure, which could offer the readers a better understanding of the presented novelty.

2.      For sections 2 to 4, lengthy discussions focus on sensor type, position, and data processing. However, most of them can be utilized in all kinds of systems. How the presented well-known technologies or methodologies can be utilized for marine structures can be introduced in detail.

3.      The organization for the review paper is proper, but the writing can be improved to provide a higher quality comprehension. For example, some abbreviations have no explanation when they firstly occur, which may lead to confusion (Like AC/DC in Table 1 and LSBG in Table 2).

4.      It included too much knowledge of SHM that is not specific to marine structures and needs to be shortened. For example, lines 417-421 present a fundamental data processing step, which seems to have little relationship with the innovation for this paper.

5.      Although review paper sometimes contains fewer figures, I would suggest adding more figures to make the paper more illustrative and easier to understand.

6.      It is recommended to include recent relevant articles on similar topics, such as Structural Concrete, 2020, 21:1272-1285; Structural Health Monitoring, 2021, 20(4): 1353-1372; Mechanical Systems and Signal Processing, 2023, 189: 110100.

Author Response

Reviewer #1:

  1. The authors claimed that the main novelty of this paper is a comprehensive review of sensors in marine structures. So, the main problem is that the authors should focus more on the difference between SHM techniques for normal infrastructures and the marine structure, which could offer the readers a better understanding of the presented novelty.

Thank you for your comment. The aim of our research is to discuss the different alternatives and options regarding structural health monitoring of marine structures. We consider very interesting what you are proposing, but the paper is already extensive enough focusing the study only on the marine industry. In future stages of the research, we will include your recommendation to compare normal infrastructures and the marine structures. We are convinced that the novelty of this paper is to collect as much information as possible in terms of the sensors and the way they are arranged in the marine structures.

  1. For sections 2 to 4, lengthy discussions focus on sensor type, position, and data processing. However, most of them can be utilized in all kinds of systems. How the presented well-known technologies or methodologies can be utilized for marine structures can be introduced in detail.

Thank you for your indication. We consider that Section 2 presents the different technologies present similar to the research put in the reference [63]. This section is very important as it introduces the reader to the subject and gives an order of magnitude of the amount of technologies present and available. Section 3 aims to give the reader an idea of the importance of optimising the location of the sensors, importance that we reflect with the amount of research related to this aspect and translated in terms of the text of section 3. We agree that most of them can be used in all types of systems but that does not mean they should be removed from this manuscript, leaving the text as it is we believe it is more understandable to the reader and strengthens the research results. With Section 4, we intend to provide information on previous research related to the processing of these signals, which has also been the subject of study on many occasions, however we agree with you that this section can be somewhat extensive and we have reduced this section by eliminating some lines taking advantage of your comment no. 4.

  1. The organization for the review paper is proper, but the writing can be improved to provide a higher quality comprehension. For example, some abbreviations have no explanation when they firstly occur, which may lead to confusion (like AC/DC in Table 1 and LSBG in Table 2).

Thank you for your observation. We have established a logical structure that allows an easy and intuitive reading, we have organised it in this way; In section 2, we establish the existing technology within each of the types pf sensors that may be available on board, with their theoretical foundations, applications and with their advantages and disadvantages. In section 3, we determine the different current theories to establish the optimal position of the sensors on board. In section 4, we review the different methodologies related to the processing of the data collected by the sensors. In section 5, we perform a historical review of the different sensors located depending on the type of ship and offshore platform. Finally, in section 6 we give the conclusions and the future trend of this present and future technology. We agree with you that both abbreviations should be specified and we have included the explanation on lines 274 and 275. On the other hand, the abbreviation for LBSG was misspelled in the table and we have corrected it, the explanation of this abbreviation is on lines 143 and 144.

  1. It concluded too much knowledge of SHM that is not specific to marine structures and needs to be shortened. For example, lines 417-421 present a fundamental data processing step, which seems to have little relationship with innovation for this paper.

Thank you for your comment. Following on from the reply given to your comment no. 2, we agree with you and what you propose and have removed the paragraph relating to those lines and have started with another heading to make this section more understandable and less lengthy.

  1. Although review sometimes contains fewer figures, I would suggest adding more figures to make the paper more illustrative and easier to understand.

Thank you for your indication. We agree with you and what you say is very interesting, however, as the paper is quite extensive we believe that putting more figures would increase the number of pages of the article without providing more information than that already shown in it. We believe that with the references we have included in the paper, the reader can find the information on the figures they are looking for with relative ease.

  1. It is recommended to include recent relevant articles on similar topics, Structural Concrete, 2020, 21: 1272-1285; Structural Health Monitoring, 2021, 20(4): 1353-1372; Mechanical Systems and Signal Processing, 2023, 189: 110100.

Thank you for your observation. We have included the references you propose and numbered them in the list of references (171, 172 and 173).

  1. Shan, J.; Zhang, H.; Shi, W.; Lu, X. Health monitoring and field-testing of high-rise buildings: A review. Structural Concrete. 2020, 21, 1272-1285.
  2. Bao Y.; Li, H. Machine learning paradigm for structural health monitoring. Structural Health Monitoring. 2021, 20(4), 1353-1372.
  3. Shan, J.; Zhuang, C.; Loong, C. N. Parametric identification of Timoshenko-beam model for shear-wall structures using monitoring data. Mechanical Systems and Signal Processing. 2023, 189, 110100.

Reviewer 2 Report

The article “Health monitoring systems for maritime structures: A review” gives a review of the state-of-the-art developments in health monitoring of marine structures.

Despite the authors claim that "they establish the different sensors with their theoretical foundations and applications to determine the different current theories to define the optimal position of the sensors on board", they do not describe all the different technologies available.

For example, they do not speak about ultrasonic guided waves like Lamb waves used in under water pipeline monitoring.

Fig 1. And 3. Are missing and there is no figure 2.

The authors should take care about their assumptions like :

-         The use of sensors for structural monitoring of marine structures is independent of the material where they have to be placed, both in steel and composite material

This is not true.

In the conclusion the following sentence should be moderated: “We have demonstrated
that the health monitoring of marine structures is a profoundly interesting area due to the information it provides for the user in the prevention of structural failure”.

The sub-paragraphs are too short, they are not useful.

Table 3 is not clear.

The English should be reviewed.

Conclusion :

In conclusion the paper needs major revisions and should be rewritten. The authors should choice and describe in details the objectives of the paper, then the structure of the article will be more clear and provides a much better understanding of the work.

Author Response

Reviewer #2:

  1. Despite the authors claim that “they establish the different sensors with their theoretical foundations and applications to determine the different current theories to define the optimal position of the sensor son board”, they do not describe all the different technologies available. For example, they do not speak about ultrasonic guided waves like Lamb waves used in under water pipeline monitoring.

Thank you for your indication. We agree with you and have therefore included guided ultrasonic waves in the manuscript under the following text "A Lamb wave represents a guided wave (GW) generated in a plate-shell assembly. Lamb waves have two basic modes: symmetric and antisymmetric. Damage prediction systems and failure prediction techniques have recently been the subject of intensive research and development in a wide range of engineering applications, such as subsea pipeline monitoring [174]". And the inclusion of a new reference:

  1. Zahoor, R.; Cerri, E.; Vallifuoco, R.; Zeni, L.; De Luca, A.; Caputo, F.; Minardo, A. Lamb Wave Detection for Structural Health Monitoring Using a Ï•-OTDR System. Sensors 2022, 22, 5962
  2. Fig. 1 and 3 are missing and there is no figure 2.

Thank you for your comment. We have removed these phrases as they refer to figures that were in early versions of the manuscript but which we finally removed because the concept of the manuscript was better understood without them.

  1. The authors should take care about their assumptions like: “The use of sensors for structural monitoring of marine structures is independent of the material where they have to be placed, both in steel and composite material”.

Thank you for your observation. What we mean in this paragraph is that the use of sensors for health monitoring is for materials of all types (namely steel and composite), although there are some that are easier to implement. This comparison of the use of sensors in both on-board materials can be seen in detail in Section 5.

  1. In the conclusion the following sentence should be moderated: “We have demonstrated that the health monitoring of marine structures is a profoundly interesting area due to the information ir provides for the user in the prevention of structural failure”.

Thank you for your comment. We agree with you and have moderated the sentence by changing the word "demonstrated" to "shown".

  1. The subparagraphs are too short, they are not useful.

Thank you for your indication. We have modified this aspect.

  1. Table 3 is not clear.

Thank you for your comment. Table 3 shows the type of sensors used on each type of ship depending on the measurement to be collected, based on reference [69] and used to explain the type of structural monitoring on board. This summary table is adopted by one of the largest ship classification societies in the world, the American Bureau of Shipping (ABS) (https://ww2.eagle.org/en.html).

  1. The English should be reviewed.

Thank you for your observation. We have revised the English.

Round 2

Reviewer 2 Report

All comments have been taken into account.